# FIN-HERDING: COMPREHENSIVE EVALUATION FOR LLM HERDING BEHAVIOR IN FINANCE DOMAIN

## ABSTRACT

Large language models (LLMs) are increasingly deployed in financial contexts, raising critical concerns about reliability, alignment, and susceptibility to adversarial manipulation. While prior finance-related benchmarks assess LLMs' capabilities in sentiment analysis (SA), question answering (QA), and named entity recognition (NER), they are often restricted to short context and therefore fail to demonstrate LLM capacities in weighing positives vs negatives and making decisions under a long financial context, which mimics the actual investment decision situation. We introduce Fin-Herding (financial herding under long and uncertain financial context), a benchmark for evaluating LLM investment decision-making when faced with uncertainty and possible human-biased opinions. Fin-Herding includes 8868 long firm-specific analyst reports, including both negative and positive aspects of firms analyzed by sophisticated analysts with investment ratings (Bullish/Neutral/Bearish) spanning from various industries. We present large language models with firm analyst reports with/without analyst investment ratings, respectively, to get investment ratings generated by LLMs. We compare LLM investment rating with analyst rating as well as quasi-true-label based on real-time stock return. Our experimental results reveal that there is a significant increase in herding score(captures the extent to which LLMs follow analyst ratings) across models when presenting analyst ratings in context, ranging from 5% to 10%. Masking human opinions can encourage LLMs to think independently, regardless of right or wrong. We believe that Fin-Herding can advance future research in the area of automatic investment trading using an LLM agent.

## 1 INTRODUCTION

The advancements in large language models(LLMs) have gained significant attention for their application in various domains. Recent studies have studied the use of LLMs in the finance domain, such as financial sentiment analysis (Wang et al. 2023), finance question answering (Zhao et al. 2024), and stock trading (Yu et al. 2024). While the existing literature (e.g. Chen et al. 2021; 2022; Mateega et al. 2025) has provided a broad range of evaluation benchmarks in the finance domain, few of them comprehensively evaluate LLM decision-making behavior given long context. Decision-making requires identifying relevant signals, reconciling conflicting evidence, and synthesizing heterogeneous data into a coherent judgment, which represents the core intellectual skills that large language model (LLM) agents must master—information gathering, coordination, and grounded reasoning. Some studies (e.g., develop advanced LLM agent frameworks to combine and analyze financial documents from various sources, including news, financial reports, social media etc. A major problem is that most financial documents contain individual human opinions which are likely to be biased. For instance, management guidance in MD&A section of 10K tend to overestimate the firm future performance (Hribar & Yang 2016); analysts tend to provide 'bullish' rather than 'bearish' ratings for firms due to overconfidence or strategic incentives. Whether LLMs will herd to human-biased opinions still remains unresolved. If LLMs do herd human bias a lot, LLMs will fail to give the right answer when most 'people' are wrong which is not uncommon in financial market trading.

To bridge the gap, we introduce Fin-Herding, a financial investment-rating dataset comprising 8,868 input–label pairs. Each input is a long-form, firm-specific analyst report—produced from in-depth research—covering firms across multiple sectors, including Technology, Industrials, Financial Services, Healthcare, Utilities, Energy, Communication Services, and Real Estate. The

first sentence of each report explicitly states the analyst's investment rating for the covered firm (Bullish / Neutral / Bearish), for example, "We are maintaining our HOLD rating on XXX." We use Fin-Herding to study LLM herding behavior—the tendency of models to follow human opinions present in context rather than produce independent judgments. To isolate the effect of analyst cues, we create two perturbations of each report: (1) removal of the first sentence that contains the analyst rating; and (2) retention of the first sentence but replacement of the original rating with an alternative (synthetic) rating. For evaluation, we prompt LLMs with a chain-of-thought instruction and feed them three variants of each report (original, first-sentence removed, and first-sentence with a fake rating). We then compare the models' predicted investment ratings across these conditions to quantify the influence of explicit analyst signals on model decisions. We introduce **Herding Score**, which captures how much LLM ratings align with analyst ratings. The results suggest that overall, the herding score increases a lot when including analyst ratings in context. The difference ranges from 5% to 10% depending on models. Besides, we use a fine-grained quantile long-short portfolio method to get ground-truth investment rating label based on real stock market return to evaluate the LLM investment rating under different conditions. Specifically, we first group report by issue year and calculate three-month cumulative abnormal return (alpha) for each firm following the analyst report issue date, which efficiently control for firm-specific risk and year effect. Then we sort individual firm alpha and categorize the top 30% percentile as 'bullish', the lower 30% percentile as bearish and the rest as 'neutral'. The experiment results indicate that when analyst rating is present in context, all models have nearly the same performance as the analyst, achieving around around 33% accuracy in providing investment rating. We also find some models' performance exceeds analysts provided that the analyst rating is excluded in context, while some models' performance drop. All the above suggest LLM herding when facing complicated tasks that need intensive reasoning. The model herding does not follow the scaling rule, even cutting-edge complex models such as GPT-5, GPT-4 have possibilities to herd fake rating. To alleviate model herding, we propose to filter sentences that contain strong opinions out of the context based on Multi-Perspective Question Answering (MPQA) Subjectivity Lexicon. The results indicate the method can help improve model performance by 2-4 points especially for light open-source model like Qwen3-8B.

The main contribution of our work is three-fold:

- We conduct a comprehensive study of 18 models (Open Source & Private), showing that models are likely to herd human opinions even when human opinions take very small part in the context.

- We propose the Fin-Herding benchmark dataset, which contains long-form, summarized, time-sensitive and potentially biased firm analysis across different industries to evaluate LLM financial reasoning capacities.

- We propose a potential way to instruct large language models to avoid herding human opinions and improve independent thinking.

## 2 RELATED WORKS

### 2.1 LLM APPLICATIONS IN FINANCE

Financial texts is one of the most common unstructured data formats in the finance domain. There is a large strand of literature studying how to make use of large language models in performing financial textual analysis. One of major tasks is financial sentiment analysis, which aims to detect sentiment (positive, negative, or neutral) from financial texts and is used for investment decision-making. Prior studies provide limited financial analysis benchmark datasets. The most commonly used are Financial Phrase Bank (FPB) (Malo et al. 2014), FiQA-SA (Maia et al. 2018) and TSA (Cortis et al. 2017) for sentiment analysis. For stock prediction, some studies use LLMs to analyze stock news and predict stock prices (Yu et al., 2025; Xie et al., 2024). The major limitations of those datasets are as follows: first, most datasets are short-contexts-based, formed by simply concatenating a bunch of news/stock tweets. In finance domain, investors get information from a great deal of financial documents, forming a long context. The short sentences lacking specific contexts are of least helpful for evaluating large language models' capacities in assisting with making investment

decisions; Even concatenating stock news forms long context, they lack summarization and deep research, making LLMs susceptible to noise. Second, most prior stock prediction benchmarks only select less than 20 stocks that are well-known and concentrated in the technology industry (e.g. Apple, Google, Tesla etc), failing to evaluate LLM stock analysis comprehensively. Third, financial corpus contain human bias (over-optimism/over-pessimism), however, previous benchmarks have not uncovered LLMs' capacities in correcting human bias in the finance domain. Therefore, how do large language models weight positives versus negatives and make rational decision faced with long contexts is still unresolved in finance domain. Our paper fills in this gap by comprehensively evaluating LLM capability in correcting human bias and making rational investment decisions given long and synthesized contexts for a wide range of stocks from across different industries.

## 2.2 LLM STRATEGIC DECISION-MAKING

Strategic decision-making (Punt 2017) evaluates the model's proficiency in synthesizing diverse information to formulate and implement trading strategies, a challenge even for experts. A bunch of studies provides evaluation on LLMs' decision-making behavior under uncertainty. For instance, LLMs generally exhibit patterns similar to humans, such as risk aversion and loss aversion, with a tendency to overweight small probabilities (Jia et al. 2024). Liu et al. 2024 evaluate how LLMs perform in a financial investing situation with a variety of uncertainties, however, they only use historical prices as the context. In contrast, we use long plain financial texts to form context. Prior literature (e.g. Lyons et al., 2021; Bosquet et al., 2015) suggest that human bias (e.g. overconfidence) may be a crucial factor for explaining how false and low-quality information spreads via news, social media, firm report etc. However, most LLM agent for trading frames (e.g. Xiao et al. 2024) rely on those biased texts without comprehensively evaluating to what extent LLMs can correct human bias hidden in the contexts. Huang et al. 2019 suggests the existence of sentiment bias which can pose a concern for using the text generated by language models in downstream applications.

## 3 DATASET CONSTRUCTION

**Data Collection.** We collect analyst reports with investment ratings sourced from Yahoo Finance. We randomly download reports from different sectors, including Technology, Industrials, Financial Services, Healthcare, Utilities, Energy, Communication Services and Real Estate. The reports are in pdf format and we extract the content from pdf using large language models. The analyst reports contain analysts' deep research for a specific firm's future stock price. They collect information from different sources ranging from financial statements, MD&A to earnings conference calls and summarize multiple factors influencing company stock return, including macroeconomic situation, industry situation as well as firm fundamentals& strategy in a professional way, which is of higher quality than stock news & tweets. The reports tend to provide an investment rating(bullish/neutral/bearish) in the first sentence. The total number of analyst reports is 8868. Table 1 provides detailed statistics for each sector. The total number of reports for each sector is unbalanced but reflects the real market where most public firms lie in technology, health care and industrials. Each report contains an average of about 4,000 tokens, providing a relatively long context for large language models. An example analyst report is shown in Figure 1.

**Data Perturbation.** We evaluate the susceptibility of LLMs to human-like biases—specifically herding behavior—when making investment decisions under long-context settings by systematically perturbing the analyst reports provided to the models. In particular, we prompt LLMs to generate investment ratings based on analyst analyses that have been modified to exhibit varying degrees of bias, allowing us to assess how sensitive the models are to biased or misleading contextual cues. There is a large strand of literature (e.g. Chan et al., 2007; Grinblatt et al., 2023; Bosquet et al., 2015) indicating analysts forecasts are systematically biased upward due to overconfidence and strategic incentives (ties to investment banking). Table 2 suggests most analyst reports provide 'bullish' rating in our sample (72.28%), while very few analysts provide 'bearish' rating (0.29%), which proves the fact documented in prior literature that analysts tend to over-optimistic. To gauge different LLMs' herding behavior, we conduct two comparison experiments: first, we provide LLMs with analyst reports w/o obvious analyst ratings and compare the differences in LLM investment ratings. Although an analyst report may contain implicit biases throughout the narrative, the analyst's investment rating (e.g., buy/bullish, hold/neutral, sell/bearish) is typically stated explicitly in the opening

> **Analyst Report Sample**
>
> We are maintaining our HOLD rating on Oracle Corp. (NGS: ORCL). We have a sense of relief that the Oracle/TikTok deal is much more in line with a commercial transaction than a merger. Rumors swirling around the deal at one point had Oracle acquiring TikTok. In the actual deal, Oracle locks in TikTok's growing business for its small Oracle Cloud service as it gets a stake in a high-growth asset which could one day file for an IPO. Our biggest fears were around Oracle acquiring ownership in a business that had no strategic fit in an area, social media, in which it has zero experience. While it is doubtful that the deal will actually accomplish true security when it comes to TikTok's influence on U.S. social media, we think Oracle should be capable of fulfilling its commitments on U.S. user data security.
>
> ....
>
> Oracle Corp. is the world's largest independent enterprise software company, with annual revenues of $39 billion. Its software products include database, middleware, application and cloud-based software designed for general business purposes and for specific industries. In addition, Oracle provides product upgrades, maintenance releases and patches through license update agreements. It also offers product support through the cloud, internet and global support centers. Oracle also provides server hardware through its acquisition of Sun Microsystems in 2010.

Figure 1: A sample analyst report used as input to the model.

sentence. Therefore, in our first experiment, we perturb the original report by removing only this initial sentence that conveys the explicit investment recommendation. This allows us to examine the extent to which the LLM relies on human-provided views when forming its own investment decision; Second, to further evaluate LLM herding behavior, we introduce a fabricated analyst rating by deliberately altering the rating stated in the first sentence of the original report. This enables us to assess the extent to which LLMs adopt—or "herd toward"—a false rating even when it contradicts the subsequent analysis.

**Ground-truth labels.** To evaluate the practical implications of LLM herding—specifically, whether alignment with human opinions improves or degrades investment decisions—it is necessary to obtain ground-truth investment ratings. Because analyst-issued investment ratings are known to exhibit substantial and systematic bias, they are unsuitable as ground-truth labels for empirical prediction tasks. Consequently, we rely on realized stock returns to define true outcomes. Prior studies (e.g., Liu et al. 2023; Yu et al. 2024) typically construct labels using daily returns combined with ad-hoc threshold rules—for example, designating returns above 1% as "bullish," between -1% and 1% as "neutral," and below -1% as "bearish." However, such cutoffs are inherently arbitrary and may not accurately control for firm-specific risk and capture economically meaningful variation in firm performance. To address this limitation, we develop a fine-grained approach for constructing ground-truth investment ratings using portfolio-based alpha generation standard. Specifically, we employ a quantile-based portfolio classification derived from the three-month (approximately 60 trading days) cumulative abnormal return (CAR) following the report issuance date. This design reflects the fact that analyst reports primarily target a firm's medium- to long-term performance rather than short-horizon fluctuations in daily returns. On average, analysts update their forecasts when firm issuing quarterly report, so we believe 60 trading days is average forecast horizons. In particular, for each firm, given a report issued on YYYY/MM/DD, we compute the 60-day abnormal return ($\alpha$) using the market model. As shown in Equation (1), we regress firm-level returns on market returns to estimate the risk-adjusted benchmark ($\beta$), and the resulting residual captures the abnormal performance (alpha). Because investors such as hedge funds are fundamentally concerned with risk-adjusted abnormal returns rather than raw buy-and-hold performance, cumulative abnormal return provides a more economically meaningful basis for defining true investment ratings. After computing each report's 60-day cumulative abnormal return, we classify investment outcomes using a year-specific, quantile-based approach inspired by long-short portfolio method widely documented in finance literature. For all reports issued in year t, we sort the post-report cumulative abnormal returns and assign ratings based on their relative positions within the annual distribution. Specifically, observations in the upper 30% quantile are labeled bullish, those in the lower 30% quantile are labeled bearish, and

Table 1: Sample Information

| Sector | Num | Percentage |
|---|---|---|
| Technology | 1491 | 16.8% |
| Industrials | 1436 | 16.2% |
| Financial Services | 1367 | 15.4% |
| Healthcare | 1288 | 14.5% |
| Consumer Defensive | 982 | 11.07% |
| Utilities | 701 | 7.9% |
| Communication Services | 590 | 6.6% |
| Energy | 615 | 6.9% |
| Real Estate | 398 | 4.5% |
| Total | 8868 | 100% |

the remaining middle 40% are categorized as neutral. This procedure ensures that classifications are comparable across years and are not distorted by time-varying market conditions.

$$R_{it} = \beta r_{mt} + \alpha_{it} \tag{1}$$

The second half of table 2 shows that, following analyst report date, based on our quantile-based investment rating classification method, approximately 30% of stocks fall into the bullish category, roughly 30% into the bearish category, and the remaining 40% into the neutral category, indicating a relatively balanced distribution across years. More importantly, this distribution is substantially more balanced than that observed in analyst-issued ratings.

Table 2: Sample Investment Rating

| Analyst Investment Rating | Num | Percentage |
|---|---|---|
| Bullish | 6410 | 72.28% |
| Bearish | 26 | 0.29% |
| Neutral | 2432 | 24.74% |
| Total | 8868 | 100% |
| Return-based Rating | | |
| Bullish | 2664 | 30% |
| Bearish | 2662 | 30% |
| Neutral | 3542 | 40% |

## 4 EXPERIMENTS

### 4.1 EXPERIMENT SETUP

**Models.** We evaluate a diverse set of language models to capture a broad spectrum of capabilities, spanning both open-source and closed-source models. Specifically, we include: GPT-5.1 (OpenAI 2025), GPT-4.1 (OpenAI 2024), Claude-3.5-Haiku (Anthropic 2024), Claude-4-Sonnet (Anthropic 2025). We also include open-source models such as Meta-Llama-3-8B-Instruct(AI 2024b), Meta-Llama-3.1-8B-Instruct(AI 2024a), Mistral-7B-Instruct-v0.3(AI 2024c), Mistral-Nemo-Instruct-2407(AI 2024d), gemma-7b-instruct(DeepMind 2024c), gemma-2-9b-instruct(DeepMind 2024b), gemma-2-27b-instruct(DeepMind 2024a), Qwen2-7B-Instruct(Group 2024, Qwen2.5-7B-Instruct(Team 2024), Qwen2.5-14B-Instruct(Team 2024), Qwen3-8B (Team 2025), DeepSeek-V2-Lite-Chat (DeepSeek-AI 2024), internlm2-chat-7b (Laboratory 2024), Yi-1.5-9B-Chat(01.AI 2024), glm-4-9B-chat(THUDM 2024).

**Metrics.** To capture how much LLMs herd to analysts/manipulated ratings when providing investment rating, we introduce **Herding Score**, which assigns 1 when a model's rating $m_i$ is as

---

**Chain-of-Thought Prompt**

**[System Input]:**
"You are an investor. Analyze the firm analyst report logically.
Then provide your own investment rating.
Format as a JSON object with the following fields:
answer: The precise answer to the question. Only one of bullish, neutral, bearish.
reason: One or more paragraphs indicating why you provide the answer.

**[User Input]:**
{analyst report content}

---

Figure 2: Prompt Template

same as the rating from analyst rating/manipulated rating $a_i$ and zero otherwise and then take average.

$$\text{Herding Score} = \frac{1}{N} \sum_{i=1}^{N} I(m_i, a_i) \tag{2}$$

To assess whether alignment with human opinions enhances or impairs investment decision-making, we compute the accuracy of LLM-generated investment ratings using our ground-truth labels. Higher accuracy reflects stronger model capability in predicting stock returns through financial reasoning under long, potentially biased textual inputs. The accuracy differential between predictions made with and without embedded analyst ratings captures the extent to which herding behavior influences LLM decisions. The prompt template used in this evaluation is presented in Figure 2.

## 4.2 RESULTS

Table 3 reveals the herding behavior for each model under two different conditions: with or without analyst rating in context. For each cell, the left hand side of '/' is herding score under the case where analyst rating is included in report, the right hand side of '/' is herding score under the case where analyst rating is removed. We find that the inclusion of analyst ratings significantly strengthens model herding behavior across nearly all categories and models. Specifically, the results suggest that when analyst ratings are not explicitly stated, private models (like GPT-5, GPT-4, and Claude variants) generally align with analyst ratings more than open-source models, with average herding scores in the mid-80s to low-90s. For instance, GPT-5 achieves an average of 87.4%, while GPT-4 follows closely at 89.5%. Open-source models, by contrast, display more variability, with averages ranging from the mid-60s (e.g., Yi-1.5-9b-Chat at 69.7%) to the mid-80s (e.g., Llama-3-8b-it at 84.4%). This is intuitive because private models such as GPT, Claude tend to have way more parameters to capture contextual and semantic information, making them align with analysts better even without explicit rating guidance. In contrast, table 3 also shows a striking shift once analyst ratings are provided. Across both private and open-source models, herding scores rise sharply, with many models reaching averages above 90%. GPT-5 and GPT-4, for instance, jump to 94.6% and 95.9%, respectively, while even mid-performing open-source models like gemma-7b-it and Mistral-7B-it-v0.3 achieve scores exceeding 96%. Notably, models that previously lagged in case of no analyst rating (such as Yi-1.5-9b-Chat or Qwen2-7b-it) demonstrate significant improvements, often surpassing 90% in most categories. This suggests that human ratings act as a strong guiding signal, discouraging models' independent thinking and reducing divergence in model outputs and encouraging greater alignment across sectors. One noticeable thing is that the increased model herding behavior from adding explicit analyst ratings are not uniform across sectors. For example, in table 3, sectors like Financial Services and Healthcare already see relatively high alignment (often above 85–90%), while sectors like Communication Services and Utilities show greater variability. When analyst ratings are provided, these weaker sectors benefit disproportionately, with scores consolidating above 90% for nearly all models. This highlights that external expert input (analyst ratings) particularly enhances agreement in areas where model consensus is weaker.

Table 3: Model Herding w/o Analyst Rating

| Models | Communication Services | Consumer Defensive | Energy | Financial Services | Healthcare |
|---|---|---|---|---|---|
| Private Models | | | | | |
| GPT-5 | 82.7/70.2 | 93.2/88.3 | 92.3/78.9 | 95.1/87.3 | 95.3/84.1 |
| GPT-4 | 84.1/75.4 | 94.8/86.2 | 93.2/89.0 | 96.3/94.3 | 96.9/92.2 |
| Claude-3.5-Haiku | 84.5/76.4 | 95.8/90.0 | 94.4/85.4 | 96.3/91.3 | 97.9/95.3 |
| Claude-4-Sonnet | 85.7/56.7 | 93.2/81.7 | 88.3/75.0 | 90.5/84.2 | 94.1/83.3 |
| Open-source | | | | | |
| gemma-7b-it | 96.4/72.1 | 93.8/84.7 | 99.2/82.0 | 98.9/91.1 | 98.2/90.3 |
| gemma-2-9b-it | 95.6/68.9 | 97.6/92.2 | 96.1/77.3 | 97.9/90.9 | 97.3/90.7 |
| gemma-2-27b-it | 94.1/67.9 | 97.9/91.3 | 96.5/78.1 | 98.6/89.2 | 97.9/89.7 |
| Meta-Llama-3-8B-it | 87.22/66.3 | 96.81/86.5 | 94.95/75.8 | 96.17/89.5 | 96.41/86.9 |
| Meta-Llama-3.1-8B-it | 86.9/66.7 | 91.9/85.6 | 88.9/74.5 | 94.5/88.2 | 93.5/86.6 |
| Qwen2-7B-it | 93.5/68.6 | 94.1/88.9 | 89.9/75.7 | 96.6/86.9 | 95.1/88.1 |
| Qwen2.5-7B-it | 98.79/71.40 | 97.88/88.80 | 96.75/81.98 | 98.78/90.07 | 99.17/92.16 |
| Qwen2.5-14B-it | 96.35/74.91 | 97.24/89.98 | 95.74/80.32 | 96.90/89.33 | 96.93/88.92 |
| Qwen3-8B | 97.96/67.06 | 99.08/88.47 | 98.69/73.13 | 99.19/85.56 | 98.44/89.66 |
| internlm2-chat-7b | 86.5/60.9 | 87.6/79.4 | 87.8/71.8 | 89.0/78.0 | 90.0/81.4 |
| glm-4-9b-chat | 96.4/67.7 | 97.8/84.6 | 95.8/79.5 | 99.3/87.6 | 98.7/89.6 |
| Mistral-7B-it-v0.3 | 94.7/58.6 | 96.4/82.0 | 95.9/68.1 | 97.7/79.8 | 94.9/80.7 |
| Mistral-Nemo-it-2407 | 83.0/61.6 | 89.8/82.0 | 89.2/76.3 | 94.2/83.5 | 91.6/83.8 |
| DeepSeek-V2-Lite-Chat | 74.78/48.36 | 81.60/60.09 | 75.00/53.97 | 78.77/55.55 | 77.72/58.62 |
| Yi-1.5-9B-Chat-16K | 76.8/56.7 | 85.1/80.2 | 78.9/58.0 | 82.6/69.8 | 87.8/76.2 |
| **Models** | **Industrials** | **Real Estate** | **Technology** | **Utilities** | **Average** |
| Private Models | | | | | |
| GPT-5 | 94.9/89.3 | 95.0/77.1 | 92.7/85.5 | 91.9/86.3 | 94.6/87.4 |
| GPT-4 | 95.4/88.7 | 96.3/87.2 | 93.8/90.4 | 91.5/91.7 | 95.9/89.5 |
| Claude-3.5-Haiku | 96.2/93.4 | 93.2/87.1 | 95.5/93.6 | 94.4/92.4 | 97.1/91.0 |
| Claude-4-Sonnet | 93.9/83.4 | 87.4/70.5 | 91.1/81.5 | 92.9/76.5 | 90.6/79.5 |
| Open-source | | | | | |
| gemma-7b-it | 97.7/89.6 | 98.2/88.4 | 97.4/90.1 | 99.1/88.0 | 97.6/87.6 |
| gemma-2-9b-it | 97.4/91.5 | 95.9/84.1 | 95.6/83.8 | 97.1/90.9 | 97.0/87.7 |
| gemma-2-27b-it | 97.4/89.9 | 97.1/80.5 | 97.1/90.0 | 98.6/87.1 | 97.5/86.9 |
| Meta-Llama-3-8B-it | 97.55/88.1 | 93.42/80.8 | 93.93/84.0 | 96.84/85.4 | 95.37/84.4 |
| Meta-Llama-3.1-8B-it | 94.1/88.5 | 91.9/84.8 | 91.2/85.8 | 94.0/84.7 | 92.4/84.5 |
| Qwen2-7B-it | 97.1/88.4 | 92.5/84.2 | 90.4/85.0 | 97.4/87.2 | 94.3/85.1 |
| Qwen2.5-7B-it | 98.10/90.27 | 98.52/90.24 | 98.80/93.10 | 99.19/91.04 | 98.51/89.11 |
| Qwen2.5-14B-it | 95.35/88.25 | 97.70/86.5 | 94.41/91.70 | 97.81/86.86 | 96.27/87.65 |
| Qwen3-8B | 98.05/84.72 | 97.98/82.32 | 96.91/89.91 | 98.86/80.31 | 98.30/84.42 |
| internlm2-chat-7b | 90.4/81.7 | 85.9/78.7 | 88.9/74.0 | 92.2/80.7 | 89.1/77.5 |
| glm-4-9b-chat | 98.3/88.0 | 97.5/87.0 | 98.0/90.5 | 98.6/88.0 | 98.1/86.2 |
| Mistral-7B-it-v0.3 | 97.4/82.6 | 94.2/74.5 | 96.4/83.3 | 97.6/80.5 | 96.4/78.8 |
| Mistral-Nemo-it-2407 | 94.1/84.4 | 92.4/78.9 | 87.2/80.2 | 92.7/81.6 | 90.8/80.7 |
| DeepSeek-V2-Lite-Chat | 83.22/58.06 | 72.26/58.06 | 71.76/59.07 | 82.49/59.25 | 77.98/57.32 |
| Yi-1.5-9B-Chat-16K | 86.7/71.3 | 78.2/65.5 | 78.7/68.1 | 82.2/65.8 | 83.4/69.7 |

Table 4 displays experiment results for model herding score when perturbing original analyst rating statement shown at the beginning of analyst report. The true analyst rating is deliberately replaced by another different fake rating which may be contradictory to the main analysis content. The results suggest that there is a large variation in herding fake rating between models, ranging from 10% to 60%. The average herding score across all models is approximately 30%, indicating that LLMs—even state-of-the-art systems such as GPT-5 and GPT-4—remain susceptible to inheriting unsupported human biases embedded in the input. This suggests that substantial potential for model herding persists, even among the most advanced architectures. Additionally, the degree of herding is not strongly correlated with model size.

Table 5 reports LLM performance in providing stock investment ratings based solely on analyst reports using real time stock return as ground-truth label (see section 3), with or without access to the analysts' explicit investment ratings. When analyst ratings are provided, nearly all models achieve accuracy levels close to those of human analysts, indicating strong herding behavior. The average analyst accuracy is approximately 33% and the accuracy of most models falls between 32% - 34%. In contrast, when the first sentences that contain explicit analyst ratings are removed, model

Table 4: Model Herding with 'fake' rating

| Model | Comm. Services | Cons. Defensive | Energy | Financial Services | Healthcare | Industrials | Real Estate | Technology | Utilities | Avg |
|---|---|---|---|---|---|---|---|---|---|---|
| Private Models | | | | | | | | | | |
| GPT-5 | 50.00 | 39.61 | 32.20 | 26.70 | 28.49 | 29.80 | 38.64 | 32.19 | 42.65 | 33.54 |
| GPT-4 | 64.24 | 41.55 | 60.65 | 58.38 | 51.24 | 43.87 | 57.07 | 30.71 | 55.92 | 48.77 |
| Claude-3.5-Haiku | 60.23 | 39.34 | 58.65 | 54.97 | 48.12 | 46.03 | 54.62 | 36.41 | 50.71 | 45.98 |
| Claude-4-Sonnet | 52.06 | 40.60 | 37.09 | 30.75 | 33.40 | 43.73 | 35.45 | 36.70 | 40.02 | 35.85 |
| Open-source Models | | | | | | | | | | |
| Qwen2.5-7B-it | 41.33 | 22.96 | 41.19 | 33.96 | 33.01 | 34.07 | 32.96 | 14.77 | 34.08 | 30.33 |
| Qwen2-7B-it | 52.65 | 28.47 | 42.29 | 40.66 | 31.10 | 22.02 | 36.48 | 20.57 | 33.28 | 31.69 |
| Qwen2.5-14B-it | 65.20 | 38.89 | 60.79 | 62.80 | 50.56 | 33.66 | 54.55 | 35.17 | 40.00 | 46.64 |
| Qwen3-8B | 46.18 | 31.53 | 64.72 | 45.79 | 44.95 | 26.17 | 45.57 | 17.43 | 38.25 | 36.89 |
| gemma-2-27b-it | 68.76 | 43.25 | 67.64 | 62.00 | 53.81 | 42.20 | 58.22 | 23.07 | 55.51 | 48.78 |
| gemma-2-9b-it | 48.01 | 19.68 | 49.25 | 29.15 | 30.06 | 16.61 | 33.18 | 17.74 | 31.39 | 27.14 |
| gemma-7b-it | 19.93 | 11.62 | 20.98 | 7.91 | 9.95 | 12.59 | 9.34 | 7.57 | 12.98 | 11.48 |
| internlm2-chat-7b | 39.91 | 30.46 | 39.55 | 35.28 | 29.70 | 25.08 | 29.55 | 25.23 | 26.25 | 30.34 |
| Mistral-7B-it-v0.3 | 69.90 | 59.04 | 72.96 | 69.77 | 68.35 | 55.24 | 65.15 | 37.74 | 66.38 | 60.42 |
| Mistral-Nemo-it-2407 | 50.17 | 30.00 | 40.52 | 37.81 | 29.03 | 26.50 | 30.96 | 22.45 | 29.18 | 31.28 |
| Yi-1.5-9B-Chat-16K | 32.43 | 27.24 | 38.50 | 34.63 | 33.80 | 32.40 | 34.01 | 30.85 | 34.91 | 32.80 |
| DeepSeek-V2-Lite-Chat | 57.09 | 56.80 | 55.54 | 60.69 | 56.60 | 55.52 | 58.87 | 47.52 | 59.86 | 55.87 |
| Meta-Llama-3-8B-it | 64.40 | 42.78 | 50.41 | 47.95 | 48.37 | 39.40 | 46.95 | 56.06 | 47.28 | 48.57 |
| Meta-Llama-3.1-8B-it | 34.80 | 17.60 | 26.63 | 19.35 | 19.17 | 12.08 | 22.78 | 14.12 | 19.66 | 18.78 |
| glm-4-9b-chat | 70.99 | 46.61 | 62.56 | 53.73 | 53.65 | 45.29 | 51.91 | 39.88 | 58.56 | 51.30 |

accuracy diverges substantially, ranging from -6% to +2% relative to analysts. The removal of first rating sentence tend to have least effects on the performance of private models like GPT-5, GPT-4, Claude-3.5-Haiku and Claude-4-Sonnet, the change of accuracy for each model is less than 1%. In contrast, several open-source models fall notably below analyst performance, including gemma-2-9b-it (-6%), gemma-7b-it (-6%), Meta-Llama-3-8B-Instruct (-6%), Meta-Llama-3.1-8B-Instruct (-6%), internlm2-chat-7b (-6%), Yi-1.5-9B-Chat (-6%), and glm-4-9b-chat (-6%). At the same time, several models outperform analysts without access to analyst ratings, such as Mistral-Nemo-it-2407 (+1%), DeepSeek-V2-Lite-Chat (+1%), Mistral-7B-it-v0.3 (+1%), and Qwen3-8B (+1%). The large drop in accuracy when analyst ratings are removed further illustrates herding behavior: models rely heavily on human opinions to perform well. Although alignment with analyst views improves apparent accuracy for some models, it also shows that models lack independent reasoning, as their performance does not surpass human analysts. Across industries, LLMs generally achieve higher accuracy in sectors such as Financial Services, Utilities, and Technology.

## 4.3 ALLEVIATING MODEL HERDING VIA BIAS AWARENESS

We propose a method to mitigate model herding in discrete decision-making by filtering human opinions from the input context, thereby encouraging models to reason independently. Although we remove the explicit analyst rating from the first sentence of each report, the remaining text often still contains numerous sentences expressing explicit or implicit subjective judgments, which can induce herding behavior in LLMs. To identify such potentially biased statements, we employ the Multi-Perspective Question Answering (MPQA) Subjectivity Lexicon, a widely used linguistic resource for detecting opinion-bearing and subjective expressions in text Wiebe et al., 2005. The lexicon comprises several thousand lexical items annotated with rich subjectivity metadata—including polarity (positive, negative, neutral, or both), subjectivity strength ("strongsubj" or "weaksubj"), and part-of-speech categories (noun, verb, adjective, adverb). In our implementation, we perturb each analyst report by removing sentences that contain any lexical item labeled as strongsubj in MPQA. As reported in Table 6, the method proves effective for both private models and open-source models. The private models like GPT-4, Claude-4-Sonnet improves 2%, while GPT-5 only improves 0.5%. In contrast, for open-source models, the accuracy of LLM-generated investment ratings mostly improves significantly after excluding all opinionated content, with gains of approximately 2–4 percentage points relative to the case where only the first rating sentence is excluded. Most importantly, some models including Qwen3-8B, DeepSeek-V2-Lite-Chat and Meta-Llama-3-8B-It perform even better than analysts as well as cutting-edge complex models like GPT-5. These results suggest that filtering biased subjective expressions enhances LLM performance by reducing reliance on human opinions embedded in the text and light open-source models have great potentials if properly prompted which can help save costs.

Table 5: Model Accuracy with/without Analyst Rating

| Models | Communication Services | Consumer Defensive | Energy | Financial Services | Healthcare |
|---|---|---|---|---|---|
| Private Models | | | | | |
| GPT-5 | 32.99/33.22 | 34.80/35.64 | 35.11/33.33 | 31.87/32.55 | 30.09/31.13 |
| GPT-4 | 31.56/32.82 | 33.79/34.94 | 34.06/32.14 | 32.42/33.50 | 31.28/32.94 |
| Claude-3.5-Haiku | 31.13/32.39 | 34.02/34.93 | 34.30/33.39 | 31.45/34.20 | 30.10/30.83 |
| Claude-4-Sonnet | 32.46/33.05 | 33.90/34.82 | 35.20/34.60 | 33.08/34.53 | 31.09/31.88 |
| Open-source | | | | | |
| gemma-7b-it | 30.28/24.33 | 32.52/27.23 | 32.04/26.03 | 34.13/29.69 | 31.24/25.88 |
| gemma-2-9b-it | 29.89/29.09 | 33.48/30.07 | 33.39/25.85 | 35.05/28.94 | 31.50/27.69 |
| gemma-2-27b-it | 27.94/28.76 | 33.94/33.81 | 32.55/32.63 | 34.13/33.09 | 31.22/30.27 |
| Meta-Llama-3-8B-it | 32.17/26.91 | 34.17/28.76 | 31.29/28.11 | 34.14/28.41 | 31.13/27.07 |
| Meta-Llama-3.1-8B-it | 33.55/25.36 | 33.33/29.12 | 34.83/29.72 | 34.25/29.02 | 31.36/25.94 |
| Qwen2-7B-it | 30.44/26.28 | 34.59/28.14 | 32.23/26.38 | 33.81/27.83 | 30.72/25.82 |
| Qwen2.5-7B-it | 28.96/33.59 | 33.37/32.52 | 32.21/31.12 | 33.19/33.49 | 31.23/31.38 |
| Qwen2.5-14B-it | 30.98/34.76 | 31.69/35.31 | 34.01/33.74 | 32.23/33.14 | 31.39/30.53 |
| Qwen3-8B | 29.37/33.88 | 33.91/34.50 | 31.86/31.12 | 33.82/35.06 | 31.70/32.61 |
| internlm2-chat-7b | 30.70/24.85 | 36.01/26.58 | 33.49/28.17 | 34.53/29.32 | 30.45/25.49 |
| glm-4-9b-chat | 30.08/24.90 | 32.76/26.25 | 32.95/26.90 | 34.28/28.74 | 31.29/24.71 |
| Mistral-7B-it-v0.3 | 30.41/35.04 | 33.43/33.53 | 33.01/32.85 | 34.13/34.06 | 30.72/32.67 |
| Mistral-Nemo-it-2407 | 31.73/33.60 | 34.80/35.16 | 33.92/31.07 | 34.37/34.47 | 30.98/33.40 |
| DeepSeek-V2-Lite-Chat | 32.02/34.83 | 34.05/35.59 | 32.13/32.40 | 35.12/35.46 | 33.88/32.01 |
| Yi-1.5-9B-Chat-16K | 32.85/28.45 | 36.19/29.46 | 34.67/28.53 | 36.53/30.35 | 32.90/28.12 |
| Analyst | 28.98 | 33.81 | 32.36 | 33.80 | 31.60 |

| Models | Industrials | Real Estate | Technology | Utilities | Average |
|---|---|---|---|---|---|
| Private Models | | | | | |
| GPT-5 | 31.22/33.98 | 28.66/28.78 | 33.24/34.67 | 35.05/37.37 | 32.98/33.60 |
| GPT-4 | 30.90/33.13 | 28.93/29.94 | 33.03/33.59 | 33.13/35.89 | 33.05/33.87 |
| Claude-3.5-Haiku | 30.88/32.46 | 27.57/27.94 | 32.80/33.09 | 32.52/34.47 | 32.58/33.45 |
| Claude-4-Sonnet | 30.91/33.90 | 29.13/29.56 | 32.5/34.50 | 34.98/35.84 | 33.04/33.78 |
| Open-source | | | | | |
| gemma-7b-it | 31.75 / 27.90 | 30.50 / 24.90 | 36.58 / 26.31 | 32.97 / 24.16 | 32.93 / 26.69 |
| gemma-2-9b-it | 31.70 / 28.36 | 32.68 / 27.97 | 36.78 / 26.13 | 32.07 / 25.89 | 33.02 / 27.84 |
| gemma-2-27b-it | 31.36 / 31.15 | 32.14 / 30.72 | 36.06 / 37.15 | 32.10 / 32.57 | 32.56 / 32.28 |
| Meta-Llama-3-8B-it | 32.21 / 28.38 | 32.08 / 26.86 | 37.04 / 26.36 | 32.45 / 26.03 | 33.35 / 27.48 |
| Meta-Llama-3.1-8B-it | 31.76 / 28.13 | 31.75 / 25.10 | 36.57 / 26.26 | 32.32 / 24.49 | 33.47 / 27.19 |
| Qwen2-7B-it | 32.12 / 28.87 | 30.26 / 25.73 | 36.19 / 26.76 | 34.04 / 25.92 | 33.13 /27.08 |
| Qwen2.5-7B-it | 32.11 /32.80 | 31.34 /33.71 | 35.39 /36.62 | 33.28 /31.21 | 32.70/33.21 |
| Qwen2.5-14B-it | 33.20 / 32.40 | 31.49 / 30.84 | 33.15 / 35.09 | 30.91 / 31.66 | 32.28 / 33.14 |
| Qwen3-8B | 32.19 / 33.54 | 31.06 / 30.75 | 36.75 / 37.44 | 32.95 / 33.52 | 33.14 / 34.15 |
| internlm2-chat-7b | 32.16 / 29.03 | 31.45 / 26.40 | 36.14 / 26.45 | 32.75 / 24.87 | 33.39 / 27.05 |
| glm-4-9b-chat | 32.42 / 27.91 | 30.15 / 25.26 | 36.89 / 26.57 | 31.62 / 23.37 | 33.07 / 26.38 |
| Mistral-7B-it-v0.3 | 32.03 / 33.49 | 29.50 / 32.75 | 37.24 / 37.48 | 32.69 / 32.55 | 33.12 / 34.10 |
| Mistral-Nemo-it-2407 | 32.35 / 33.95 | 30.90 / 32.91 | 37.76 / 37.52 | 32.60 / 31.86 | 33.69 / 34.26 |
| DeepSeek-V2-Lite-Chat | 32.56/ 33.87 | 33.42 / 38.73 | 35.08 / 34.90 | 35.12 / 36.03 | 33.92/ 34.56 |
| Yi-1.5-9B-Chat-16K | 33.31 / 30.02 | 35.11 / 26.39 | 33.11 / 24.09 | 35.04 / 28.84 | 34.50 / 28.64 |
| Analyst | 31.96 | 31.06 | 36.49 | 32.95 | 33.08 |

## 5 CONCLUSION

In this study, we present a comprehensive investigation into the decision-making behavior of large language models (LLMs) in the context of financial investment analysis. By introducing the Fin-Herding benchmark, we evaluate whether LLMs exhibit herding behavior when exposed to potentially biased human opinions embedded in long-form analyst reports. Our empirical results demonstrate that LLMs are indeed susceptible to herding, with significantly higher alignment to analyst ratings when such ratings are present in the input. Notably, this alignment does not necessarily translate into better investment performance, and in some cases, models outperform analysts only when the analyst opinion is masked. These findings challenge the assumption that larger or more advanced models inherently reason more independently, and highlight the importance of carefully curating inputs to encourage unbiased and grounded financial reasoning. Overall, our work underscores the need for future LLM development to prioritize independent judgment over mere reflection of dominant human narratives, especially in high-stakes, opinion-driven domains such as finance.

Table 6: Model Accuracy without human opinions

| Model | Comm. Services | Cons. Defensive | Energy | Financial Services | Healthcare | Industrials | Real Estate | Technology | Utilities | Avg |
|---|---|---|---|---|---|---|---|---|---|---|
| Private Models | | | | | | | | | | |
| GPT-5 | 33.56 | 37.78 | 34.86 | 32.92 | 31.52 | 33.91 | 33.59 | 34.74 | 35.66 | 34.16 |
| GPT-4 | 36.27 | 36.56 | 33.82 | 34.38 | 35.87 | 35.65 | 36.11 | 36.01 | 39.37 | 35.88 |
| Claude-3.5-Haiku | 35.31 | 37.12 | 35.04 | 33.43 | 30.98 | 33.74 | 35.25 | 35.03 | 34.61 | 34.99 |
| Claude-4-Sonnet | 35.79 | 37.54 | 33.96 | 35.13 | 34.69 | 35.63 | 36.86 | 35.47 | 38.98 | 35.70 |
| Qwen2-7B-It | 32.63 | 33.12 | 34.79 | 33.03 | 31.84 | 33.12 | 38.30 | 36.57 | 33.14 | 33.82 |
| gemma-2-9b-it | 38.11 | 35.77 | 34.27 | 34.52 | 33.13 | 32.68 | 35.78 | 36.03 | 35.35 | 34.71 |
| gemma-7b-it | 33.56 | 31.87 | 32.20 | 35.04 | 31.08 | 32.17 | 35.86 | 35.77 | 29.81 | 33.12 |
| glm-4-9b-chat | 33.28 | 31.27 | 34.81 | 32.45 | 30.42 | 31.50 | 35.03 | 35.13 | 30.55 | 32.53 |
| Mistral-Nemo-It-2407 | 36.50 | 33.50 | 33.17 | 32.74 | 33.83 | 34.64 | 36.80 | 35.89 | 36.29 | 34.58 |
| Mistral-7B-It-v0.3 | 38.81 | 34.05 | 36.64 | 35.26 | 37.89 | 34.84 | 38.38 | 35.33 | 36.23 | 35.99 |
| DeepSeek-V2-Lite-Chat | 37.09 | 32.17 | 35.07 | 33.26 | 36.29 | 38.06 | 35.04 | 35.20 | 35.24 | 35.30 |
| Meta-Llama-3-8B-It | 36.56 | 35.20 | 35.02 | 35.19 | 33.33 | 33.87 | 38.07 | 34.81 | 35.71 | 34.91 |
| Meta-Llama-3.1-8B-It | 33.05 | 32.24 | 33.82 | 33.04 | 31.05 | 31.48 | 32.06 | 33.71 | 33.43 | 32.58 |
| internlm2-chat-7b | 36.01 | 32.76 | 36.54 | 32.54 | 32.70 | 33.10 | 34.49 | 35.25 | 32.09 | 33.69 |
| gemma-2-27b-it | 36.94 | 35.97 | 35.66 | 35.73 | 35.36 | 35.09 | 42.09 | 36.96 | 36.52 | 36.23 |
| Qwen2.5-14B-It | 32.59 | 33.89 | 34.28 | 35.92 | 35.37 | 32.17 | 38.46 | 35.63 | 33.67 | 34.60 |
| Qwen2.5-7B-It | 33.59 | 31.77 | 34.19 | 33.66 | 33.39 | 33.49 | 36.13 | 34.93 | 32.32 | 33.64 |
| Yi-1.5-9B-Chat-16K | 36.44 | 36.30 | 35.77 | 36.28 | 39.98 | 37.61 | 40.66 | 35.73 | 43.22 | 37.67 |
| Qwen3-8B | 36.39 | 35.27 | 36.72 | 35.48 | 37.02 | 35.59 | 39.90 | 35.91 | 40.20 | 36.49 |
| Analyst | 28.98 | 33.81 | 32.36 | 33.80 | 31.60 | 31.96 | 31.06 | 36.49 | 32.95 | 33.08 |

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

## A    THE USE OF LLMs

Our use of LLMs such as OpenAI GPT-5 is only for grammar and style editing, not for research, idea generation, or analysis. All suggested edits were reviewed for factual accuracy and faithfulness to our original text.

