# OpenReview forum: "Fin-Herding: Comprehensive Evaluation for LLM Herding Behavior in Finance Domain"
_ICLR.cc/2026/Conference — ICLR 2026 Conference Withdrawn Submission_

### Official Review · Reviewer_GmPJ · 2025-10-15

**Soundness:** 1
**Presentation:** 1
**Contribution:** 2
**Rating:** 2
**Confidence:** 4

**Summary:**

This paper introduces Fin-Herding, a benchmark designed to evaluate herding behaviour of large language models (LLMs) in financial decision-making. The dataset consists of 8,868 long analyst reports across multiple industries, each labelled with investment ratings (bullish, neutral, bearish). The authors test 18 open-source and private models under two conditions: with and without the analyst’s rating included in the input. They define a simple Herding Score to measure how often model predictions align with analysts’ ratings and also compare performance using one-month stock returns as quasi-true labels. Results show that models exhibit stronger alignment when analyst ratings are visible, suggesting susceptibility to human bias. The paper concludes that masking human opinions promotes more independent reasoning in LLMs.

**Strengths:**

- The topic itself is interesting and important.

- The dataset covers multiple industries and long financial context.

- The inclusion of both private and open-source models provides a broad comparative perspective.

**Weaknesses:**

- Generally, the novelty and contribution are limited. The paper mainly constructs a dataset and reports descriptive comparisons of herding behaviour with and without analyst ratings, but there is no in-depth analysis or exploration of why this happens. There’s no attempt to interpret the underlying mechanism using any interpretability tools, attention-score analysis, or other ablation studies (e.g., varying the position of analyst opinions). The study ends with surface-level statistics rather than any analytical or causal understanding.

- The results themselves are inconsistent. In most cases, including the analyst’s rating improves the model’s predictive ability, while in others the performance actually drops. Very few models outperform the human analyst without herding, yet some (such as internlm2-chat-7b) even surpass the analyst’s accuracy after herding. This inconsistency makes it hard to draw general conclusions, and the paper does not attempt to explain these contradictions or explore any cross-model patterns.

- The definition of “bullish” and “bearish” seems self-adapted and lacks evidence or reference support. The authors use a ±1% monthly threshold without justification, whereas it is typically defined around ±20% annually in financial literature. This self-defined scheme looks arbitrary and weakens the validity of the categorisation.

- The scoring scheme is also unclear. When the authors say the model’s rating is “the same as the analyst’s rating”, it’s not explained what the rating range actually is. If the ratings are on a scale like 1–100 or 1–10, it’s difficult to understand how such high herding scores (up to 90%) are possible even when the analyst ratings are masked. If the rating simply refers to "bullish", "bearish" and "neutral" (should refer to sideways I guess), it's not fine-grained enough. And as mentioned above the definitions are not well justified.

- The writing and presentation need substantial improvement. There are many missing justifications (for example, the definition of bullish and bearish mentioned above). In several places the paper uses the word “significant” without any statistical testing or confidence evidence. These are just minor examples, and there are many similar writing issues that contribute to an unprofessional style. The result presentation also needs to be improved. Currently, it requires frequent switching between pages to cross-check tables and compare models, which disrupts readability and understanding.

Minors:

- Typos on the quotation marks.

**Questions:**

- Can the authors justify the ±1% threshold and explain why it captures meaningful bullish/bearish distinctions?

- Have the authors considered using interpretability tools (e.g., attention analysis, gradient attribution) to investigate why models herd?

- Were the experiments conducted in a temporal setup?

- Is there a reproducibility statement, even though it is not mandatory?

---

> ### Author Response · Authors · 2025-11-27
> **Thanks for your careful review and we have made major revisions based on your comments.**
>
> Thanks for your careful review and we have made major revisions based on your comments.
>
> We want to emphasize that the paper mainly focuses on comprehensive evaluation of a broad range of models rather than imterpretability. Some model weights are not open to public, so how can we calculate their attention scores?
>
> Regarding your first comment, we have to clarify that herding score is the alignment between llm’s output and a potential biased/false opinion shown in context. It is the percentage of samples that have herding answers.  Although the herding score is simple, it is straight-forward and can be used in different situations. To further illustrate that, we add another experiment. We introduce a fabricated analyst rating by deliberately altering the rating stated in the first sentence of the original report. This enables us to assess the extent to which LLMs adopt—or “herd toward”—a false rating even when it contradicts the subsequent analysis. The results suggest that models have different degrees of “herding” rather than alignment. Even for GPT-4, there is a near 50% percentage of samples providing investment rating herding towards fake rating. The detail results are as follows:
>
> **Table: Model Herding with 'fake' rating**
>
> | Model | Comm. Services | Cons. Defensive | Energy | Financial Services | Healthcare | Industrials | Real Estate | Technology | Utilities | Avg |
> |-------|----------------|-----------------|--------|---------------------|------------|-------------|-------------|------------|-----------|------|
> | **Private Models** |||||||||||
> | GPT-5 | 50.00 | 39.61 | 32.20 | 26.70 | 28.49 | 29.80 | 38.64 | 32.19 | 42.65 | 33.54 |
> | GPT-4 | 64.24 | 41.55 | 60.65 | 58.38 | 51.24 | 43.87 | 57.07 | 30.71 | 55.92 | 48.77 |
> | Claude-3.5-Haiku | 60.23 | 39.34 | 58.65 | 54.97 | 48.12 | 46.03 | 54.62 | 36.41 | 50.71 | 45.98 |
> | Claude-4-Sonnet | 52.06 | 40.60 | 37.09 | 30.75 | 33.40 | 43.73 | 35.45 | 36.70 | 40.02 | 35.85 |
> | **Open-source Models** |||||||||||
> | Qwen2.5-7B-it | 41.33 | 22.96 | 41.19 | 33.96 | 33.01 | 34.07 | 32.96 | 14.77 | 34.08 | 30.33 |
> | Qwen2-7B-it | 52.65 | 28.47 | 42.29 | 40.66 | 31.10 | 22.02 | 36.48 | 20.57 | 33.28 | 31.69 |
> | Qwen2.5-14B-it | 65.20 | 38.89 | 60.79 | 62.80 | 50.56 | 33.66 | 54.55 | 35.17 | 40.00 | 46.64 |
> | Qwen3-8B | 46.18 | 31.53 | 64.72 | 45.79 | 44.95 | 26.17 | 45.57 | 17.43 | 38.25 | 36.89 |
> | gemma-2-27b-it | 68.76 | 43.25 | 67.64 | 62.00 | 53.81 | 42.20 | 58.22 | 23.07 | 55.51 | 48.78 |
> | gemma-2-9b-it | 48.01 | 19.68 | 49.25 | 29.15 | 30.06 | 16.61 | 33.18 | 17.74 | 31.39 | 27.14 |
> | gemma-7b-it | 19.93 | 11.62 | 20.98 | 7.91 | 9.95 | 12.59 | 9.34 | 7.57 | 12.98 | 11.48 |
> | internlm2-chat-7b | 39.91 | 30.46 | 39.55 | 35.28 | 29.70 | 25.08 | 29.55 | 25.23 | 26.25 | 30.34 |
> | Mistral-7B-it-v0.3 | 69.90 | 59.04 | 72.96 | 69.77 | 68.35 | 55.24 | 65.15 | 37.74 | 66.38 | 60.42 |
> | Mistral-Nemo-it-2407 | 50.17 | 30.00 | 40.52 | 37.81 | 29.03 | 26.50 | 30.96 | 22.45 | 29.18 | 31.28 |
> | Yi-1.5-9B-Chat-16K | 32.43 | 27.24 | 38.50 | 34.63 | 33.80 | 32.40 | 34.01 | 30.85 | 34.91 | 32.80 |
> | DeepSeek-V2-Lite-Chat | 57.09 | 56.80 | 55.54 | 60.69 | 56.60 | 55.52 | 58.87 | 47.52 | 59.86 | 55.87 |
> | Meta-Llama-3-8B-it | 64.40 | 42.78 | 50.41 | 47.95 | 48.37 | 39.40 | 46.95 | 56.06 | 47.28 | 48.57 |
> | Meta-Llama-3.1-8B-it | 34.80 | 17.60 | 26.63 | 19.35 | 19.17 | 12.08 | 22.78 | 14.12 | 19.66 | 18.78 |
> | glm-4-9b-chat | 70.99 | 46.61 | 62.56 | 53.73 | 53.65 | 45.29 | 51.91 | 39.88 | 58.56 | 51.30 |

---

> > ### Author Response · Authors · 2025-11-27
> >
> > We develop a fine-grained approach for constructing ground-truth investment ratings using portfolio-based alpha generation standard. Specifically, we employ a quantile-based portfolio classification derived from the three-month (approximately 60 trading days) cumulative abnormal return (CAR) following the report issuance date. This design reflects the fact that analyst reports primarily target a firm’s medium- to long-term performance rather than short-horizon fluctuations in daily returns. On average, analysts update their forecasts when firm issuing quarterly report, so we believe 60 trading days is average forecast horizons. In particular, for each firm, given a report issued on YYYY/MM/DD, we compute the 60-day abnormal return ($\alpha$) using the market model. As shown in Equation (1), we regress firm-level returns on market returns to estimate the risk-adjusted benchmark ($\beta$), and the resulting residual captures the abnormal performance (alpha). Because investors such as hedge funds are fundamentally concerned with risk-adjusted abnormal returns rather than raw buy-and-hold performance, cumulative abnormal return provides a more economically meaningful basis for defining true investment ratings. After computing each report’s 60-day cumulative abnormal return, we classify investment outcomes using a year-specific, quantile-based approach inspired by long-short portfolio method widely documented in finance literature. For all reports issued in year t, we sort the post-report cumulative abnormal returns and assign ratings based on their relative positions within the annual distribution. Specifically, observations in the upper 30\% quantile are labeled bullish, those in the lower 30\% quantile are labeled bearish, and the remaining middle 40\% are categorized as neutral. This procedure ensures that classifications are comparable across years and are not distorted by time-varying market conditions. We have re-calculated all the model accuracies based on new ground-truth labels. Based on the new distribution of labels, the random guess accuracy is 33%. The stock prediction task is a very difficult task, even analyst (human) can only achieve 33% which is near random guess. A major finding of our paper is some open-source models can achieve accuracy higher than analysts after excluding all human-related opinions in the context, suggesting that resolving herding issue is beneficial for this task. The updated model accuracy is shown below.
> >
> > **Table: Model Accuracy without human opinions**
> >
> > | Model | Comm. Services | Cons. Defensive | Energy | Financial Services | Healthcare | Industrials | Real Estate | Technology | Utilities | Avg |
> > |-------|----------------|-----------------|--------|---------------------|------------|-------------|-------------|------------|-----------|------|
> > | **Private Models** |||||||||||
> > | GPT-5 | 33.56 | 37.78 | 34.86 | 32.92 | 31.52 | 33.91 | 33.59 | 34.74 | 35.66 | 34.16 |
> > | GPT-4 | 36.27 | 36.56 | 33.82 | 34.38 | 35.87 | 35.65 | 36.11 | 36.01 | 39.37 | 35.88 |
> > | Claude-3.5-Haiku | 35.31 | 37.12 | 35.04 | 33.43 | 30.98 | 33.74 | 35.25 | 35.03 | 34.61 | 34.99 |
> > | Claude-4-Sonnet | 35.79 | 37.54 | 33.96 | 35.13 | 34.69 | 35.63 | 36.86 | 35.47 | 38.98 | 35.70 |
> > | **Open-source Models** |||||||||||
> > | Qwen2-7B-It | 32.63 | 33.12 | 34.79 | 33.03 | 31.84 | 33.12 | 38.30 | 36.57 | 33.14 | 33.82 |
> > | gemma-2-9b-it | 38.11 | 35.77 | 34.27 | 34.52 | 33.13 | 32.68 | 35.78 | 36.03 | 35.35 | 34.71 |
> > | gemma-7b-it | 33.56 | 31.87 | 32.20 | 35.04 | 31.08 | 32.17 | 35.86 | 35.77 | 29.81 | 33.12 |
> > | glm-4-9b-chat | 33.28 | 31.27 | 34.81 | 32.45 | 30.42 | 31.50 | 35.03 | 35.13 | 30.55 | 32.53 |
> > | Mistral-Nemo-It-2407 | 36.50 | 33.50 | 33.17 | 32.74 | 33.83 | 34.64 | 36.80 | 35.89 | 36.29 | 34.58 |
> > | Mistral-7B-It-v0.3 | 38.81 | 34.05 | 36.64 | 35.26 | 37.89 | 34.84 | 38.38 | 35.33 | 36.23 | 35.99 |
> > | DeepSeek-V2-Lite-Chat | 37.09 | 32.17 | 35.07 | 33.26 | 36.29 | 38.06 | 35.04 | 35.20 | 35.24 | 35.30 |
> > | Meta-Llama-3-8B-It | 36.56 | 35.20 | 35.02 | 35.19 | 33.33 | 33.87 | 38.07 | 34.81 | 35.71 | 34.91 |
> > | Meta-Llama-3.1-8B-It | 33.05 | 32.24 | 33.82 | 33.04 | 31.05 | 31.48 | 32.06 | 33.71 | 33.43 | 32.58 |
> > | internlm2-chat-7b | 36.01 | 32.76 | 36.54 | 32.54 | 32.70 | 33.10 | 34.49 | 35.25 | 32.09 | 33.69 |
> > | gemma-2-27b-it | 36.94 | 35.97 | 35.66 | 35.73 | 35.36 | 35.09 | 42.09 | 36.96 | 36.52 | 36.23 |
> > | Qwen2.5-14B-It | 32.59 | 33.89 | 34.28 | 35.92 | 35.37 | 32.17 | 38.46 | 35.63 | 33.67 | 34.60 |
> > | Qwen2.5-7B-It | 33.59 | 31.77 | 34.19 | 33.66 | 33.39 | 33.49 | 36.13 | 34.93 | 32.32 | 33.64 |
> > | Yi-1.5-9B-Chat-16K | 36.44 | 36.30 | 35.77 | 36.28 | 39.98 | 37.61 | 40.66 | 35.73 | 43.22 | 37.67 |
> > | Qwen3-8B | 36.39 | 35.27 | 36.72 | 35.48 | 37.02 | 35.59 | 39.90 | 35.91 | 40.20 | 36.49 |
> > | Analyst | 28.98 | 33.81 | 32.36 | 33.80 | 31.60 | 31.96 | 31.06 | 36.49 | 32.95 | 33.08 |

---

### Official Review · Reviewer_X6Ue · 2025-10-30

**Soundness:** 2
**Presentation:** 2
**Contribution:** 2
**Rating:** 4
**Confidence:** 4

**Summary:**

his paper introduces the Fin-Herding benchmark to evaluate if llms exhibit herding in financial analyst reports. The core problem that assessing llm independence in the presence of human bias in finance is highly relevant and timely. The data are substantial, and the experiments are extensive, encompassing a wide range of models. However, the paper suffers from several fundamental flaws that undermine the reliability of its conclusions and the value of its contributions. The primary issues are: 1) The metric for "herding" is overly simplistic and lacks connection to established financial theory; 2) There is no discussion or control for the risk of data contamination ,i.e., train-test leakage; 3) The baseline prediction accuracy of the models is so low that the discussion on mitigating herding becomes questionable from a practical utility standpoint. While the problem is important, the current methodology and narrative are not yet mature enough for acceptance.

**Strengths:**

1. Dataset construction: The creation of a dataset with 8,868 long-form analyst reports across multiple sectors, coupled with the clever analyst rating experimental design, is a solid contribution.
2. Experimental scale: The evaluation of 18 open-source and closed-source models provides comprehensive empirical evidence.

**Weaknesses:**

### Weakness
1. Oversimplified Metric for "Herding" Lacks Financial Rigor
The paper's definition of "Herding Score" is merely the alignment between the llm's output and the analyst's rating. This is insufficient from a behavioral finance perspective. True Herding research focuses on whether a group of analysts systematically disregards their own signals to follow the consensus in response to new information. The setup here is closer to measuring opinion anchoring or conformity bias. Using a simple agreement metric cannot distinguish between an llm rationally agreeing with the analyst after reasoning and one mindlessly conforming.
2. Inadequate Consideration of Data Contamination Risk
The analyst reports sourced from Yahoo Finance are public and have a very high probability of being included in the training data of many llms, especially general-purpose models. If the models have already "seen" these reports and their subsequent stock performance during training, the experiment evaluates "memorization" or "alignment to a known label" rather than "reasoning capability." This invalidates all conclusions about Herding and predictive accuracy.
3. Low Baseline Model Performance Undermines Practical Relevance
As shown in Tables 5 and 6, the prediction accuracy of most models is around 40%, only marginally better than random guessing. The F1 scores are also consistently low. This raises a critical question: these models, in their current state, simply do not possess reliable financial forecasting capability. Therefore, the practical utility of studying how to make an inaccurate model "avoid following another inaccurate analyst" is highly questionable. It resembles a "blind leading the blind" problem.

**Questions:**

Suggestions:
1. Incorporate Established Financial Metrics: The authors should integrate classic metrics from the analyst herding literature, such as Lakonishok, which measures herding based on the distribution of bullish/bearish opinions within a group of analysts.
2. Analyze llm rating changes in relation to analyst rating revisions or earnings announcements. A more robust experiment would be to present two reports with conflicting opinions and observe if the llm herds towards the first opinion while ignoring contradictory evidence in the second.
3. It is imperative for the authors to use established tools to check the overlap between their test set and major training corpora.
4. The paper must explicitly report the methodology and results of the contamination check. If significant contamination is found, the authors should create a new, clean test set, e.g., using reports published after the knowledge cutoff dates of the models, and re-run the core experiments.
5. The contribution should be more explicitly framed as revealing a socio-cognitive bias in llms, e.g., conformity to authority, rather than providing a practical solution for financial decision-making. The conclusion should temper any implication of "improving investment performance" and instead emphasize the implications for llm safety, alignment, and reliability evaluation.

---

> ### Author Response · Authors · 2025-11-27
> **Thanks for your careful review and we have made major revisions based on your comments.**
>
> Thanks for your careful review and we have made major revisions based on your comments.
>
> First, regarding your first comment, we have to clarify that herding is not just the alignment between the llm's output and the analyst's rating, it is the alignment between llm’s output and a potential biased/false opinion shown in context. To further illustrate that, we add another experiment. we introduce a fabricated analyst rating by deliberately altering the rating stated in the first sentence of the original report. This enables us to assess the extent to which LLMs adopt—or “herd toward”—a false rating even when it contradicts the subsequent analysis. The results suggest that models have different degrees of “herding” rather than alignment. Even for GPT-4, there is a near 50% percentage of samples providing investment rating herding towards fake rating. The detail results are as follows:
>
> **Table: Model Herding with 'fake' rating**
>
> | Model | Comm. Services | Cons. Defensive | Energy | Financial Services | Healthcare | Industrials | Real Estate | Technology | Utilities | Avg |
> |-------|----------------|-----------------|--------|---------------------|------------|-------------|-------------|------------|-----------|------|
> | **Private Models** |||||||||||
> | GPT-5 | 50.00 | 39.61 | 32.20 | 26.70 | 28.49 | 29.80 | 38.64 | 32.19 | 42.65 | 33.54 |
> | GPT-4 | 64.24 | 41.55 | 60.65 | 58.38 | 51.24 | 43.87 | 57.07 | 30.71 | 55.92 | 48.77 |
> | Claude-3.5-Haiku | 60.23 | 39.34 | 58.65 | 54.97 | 48.12 | 46.03 | 54.62 | 36.41 | 50.71 | 45.98 |
> | Claude-4-Sonnet | 52.06 | 40.60 | 37.09 | 30.75 | 33.40 | 43.73 | 35.45 | 36.70 | 40.02 | 35.85 |
> | **Open-source Models** |||||||||||
> | Qwen2.5-7B-it | 41.33 | 22.96 | 41.19 | 33.96 | 33.01 | 34.07 | 32.96 | 14.77 | 34.08 | 30.33 |
> | Qwen2-7B-it | 52.65 | 28.47 | 42.29 | 40.66 | 31.10 | 22.02 | 36.48 | 20.57 | 33.28 | 31.69 |
> | Qwen2.5-14B-it | 65.20 | 38.89 | 60.79 | 62.80 | 50.56 | 33.66 | 54.55 | 35.17 | 40.00 | 46.64 |
> | Qwen3-8B | 46.18 | 31.53 | 64.72 | 45.79 | 44.95 | 26.17 | 45.57 | 17.43 | 38.25 | 36.89 |
> | gemma-2-27b-it | 68.76 | 43.25 | 67.64 | 62.00 | 53.81 | 42.20 | 58.22 | 23.07 | 55.51 | 48.78 |
> | gemma-2-9b-it | 48.01 | 19.68 | 49.25 | 29.15 | 30.06 | 16.61 | 33.18 | 17.74 | 31.39 | 27.14 |
> | gemma-7b-it | 19.93 | 11.62 | 20.98 | 7.91 | 9.95 | 12.59 | 9.34 | 7.57 | 12.98 | 11.48 |
> | internlm2-chat-7b | 39.91 | 30.46 | 39.55 | 35.28 | 29.70 | 25.08 | 29.55 | 25.23 | 26.25 | 30.34 |
> | Mistral-7B-it-v0.3 | 69.90 | 59.04 | 72.96 | 69.77 | 68.35 | 55.24 | 65.15 | 37.74 | 66.38 | 60.42 |
> | Mistral-Nemo-it-2407 | 50.17 | 30.00 | 40.52 | 37.81 | 29.03 | 26.50 | 30.96 | 22.45 | 29.18 | 31.28 |
> | Yi-1.5-9B-Chat-16K | 32.43 | 27.24 | 38.50 | 34.63 | 33.80 | 32.40 | 34.01 | 30.85 | 34.91 | 32.80 |
> | DeepSeek-V2-Lite-Chat | 57.09 | 56.80 | 55.54 | 60.69 | 56.60 | 55.52 | 58.87 | 47.52 | 59.86 | 55.87 |
> | Meta-Llama-3-8B-it | 64.40 | 42.78 | 50.41 | 47.95 | 48.37 | 39.40 | 46.95 | 56.06 | 47.28 | 48.57 |
> | Meta-Llama-3.1-8B-it | 34.80 | 17.60 | 26.63 | 19.35 | 19.17 | 12.08 | 22.78 | 14.12 | 19.66 | 18.78 |
> | glm-4-9b-chat | 70.99 | 46.61 | 62.56 | 53.73 | 53.65 | 45.29 | 51.91 | 39.88 | 58.56 | 51.30 |

---

> > ### Author Response · Authors · 2025-11-27
> >
> > Second, to address your “Model accuracy and Incorporate Established Financial Metric” concern, we develop a fine-grained approach for constructing ground-truth investment ratings using portfolio-based alpha generation standard. Specifically, we employ a quantile-based portfolio classification derived from the three-month (approximately 60 trading days) cumulative abnormal return (CAR) following the report issuance date. This design reflects the fact that analyst reports primarily target a firm’s medium- to long-term performance rather than short-horizon fluctuations in daily returns. On average, analysts update their forecasts when firm issuing quarterly report, so we believe 60 trading days is average forecast horizons. In particular, for each firm, given a report issued on YYYY/MM/DD, we compute the 60-day abnormal return ($\alpha$) using the market model. As shown in Equation (1), we regress firm-level returns on market returns to estimate the risk-adjusted benchmark ($\beta$), and the resulting residual captures the abnormal performance (alpha). Because investors such as hedge funds are fundamentally concerned with risk-adjusted abnormal returns rather than raw buy-and-hold performance, cumulative abnormal return provides a more economically meaningful basis for defining true investment ratings. After computing each report’s 60-day cumulative abnormal return, we classify investment outcomes using a year-specific, quantile-based approach inspired by long-short portfolio method widely documented in finance literature. For all reports issued in year t, we sort the post-report cumulative abnormal returns and assign ratings based on their relative positions within the annual distribution. Specifically, observations in the upper 30\% quantile are labeled bullish, those in the lower 30% quantile are labeled bearish, and the remaining middle 40\% are categorized as neutral. This procedure ensures that classifications are comparable across years and are not distorted by time-varying market conditions. We have re-calculated all the model accuracies based on new ground-truth labels. Based on the new distribution of labels, the random guess accuracy is 33%. The stock prediction task is a very difficult task, even analyst (human) can only achieve 33% which is near random guess. A major finding of our paper is some open-source models can achieve accuracy higher than analysts after excluding all human-related opinions in the context, suggesting that resolving herding issue is beneficial for this task. The updated model accuracy is shown below.
> >
> > **Table: Model Accuracy without human opinions**
> >
> > | Model | Comm. Services | Cons. Defensive | Energy | Financial Services | Healthcare | Industrials | Real Estate | Technology | Utilities | Avg |
> > |-------|----------------|-----------------|--------|---------------------|------------|-------------|-------------|------------|-----------|------|
> > | **Private Models** |||||||||||
> > | GPT-5 | 33.56 | 37.78 | 34.86 | 32.92 | 31.52 | 33.91 | 33.59 | 34.74 | 35.66 | 34.16 |
> > | GPT-4 | 36.27 | 36.56 | 33.82 | 34.38 | 35.87 | 35.65 | 36.11 | 36.01 | 39.37 | 35.88 |
> > | Claude-3.5-Haiku | 35.31 | 37.12 | 35.04 | 33.43 | 30.98 | 33.74 | 35.25 | 35.03 | 34.61 | 34.99 |
> > | Claude-4-Sonnet | 35.79 | 37.54 | 33.96 | 35.13 | 34.69 | 35.63 | 36.86 | 35.47 | 38.98 | 35.70 |
> > | **Open-source Models** |||||||||||
> > | Qwen2-7B-It | 32.63 | 33.12 | 34.79 | 33.03 | 31.84 | 33.12 | 38.30 | 36.57 | 33.14 | 33.82 |
> > | gemma-2-9b-it | 38.11 | 35.77 | 34.27 | 34.52 | 33.13 | 32.68 | 35.78 | 36.03 | 35.35 | 34.71 |
> > | gemma-7b-it | 33.56 | 31.87 | 32.20 | 35.04 | 31.08 | 32.17 | 35.86 | 35.77 | 29.81 | 33.12 |
> > | glm-4-9b-chat | 33.28 | 31.27 | 34.81 | 32.45 | 30.42 | 31.50 | 35.03 | 35.13 | 30.55 | 32.53 |
> > | Mistral-Nemo-It-2407 | 36.50 | 33.50 | 33.17 | 32.74 | 33.83 | 34.64 | 36.80 | 35.89 | 36.29 | 34.58 |
> > | Mistral-7B-It-v0.3 | 38.81 | 34.05 | 36.64 | 35.26 | 37.89 | 34.84 | 38.38 | 35.33 | 36.23 | 35.99 |
> > | DeepSeek-V2-Lite-Chat | 37.09 | 32.17 | 35.07 | 33.26 | 36.29 | 38.06 | 35.04 | 35.20 | 35.24 | 35.30 |
> > | Meta-Llama-3-8B-It | 36.56 | 35.20 | 35.02 | 35.19 | 33.33 | 33.87 | 38.07 | 34.81 | 35.71 | 34.91 |
> > | Meta-Llama-3.1-8B-It | 33.05 | 32.24 | 33.82 | 33.04 | 31.05 | 31.48 | 32.06 | 33.71 | 33.43 | 32.58 |
> > | internlm2-chat-7b | 36.01 | 32.76 | 36.54 | 32.54 | 32.70 | 33.10 | 34.49 | 35.25 | 32.09 | 33.69 |
> > | gemma-2-27b-it | 36.94 | 35.97 | 35.66 | 35.73 | 35.36 | 35.09 | 42.09 | 36.96 | 36.52 | 36.23 |
> > | Qwen2.5-14B-It | 32.59 | 33.89 | 34.28 | 35.92 | 35.37 | 32.17 | 38.46 | 35.63 | 33.67 | 34.60 |
> > | Qwen2.5-7B-It | 33.59 | 31.77 | 34.19 | 33.66 | 33.39 | 33.49 | 36.13 | 34.93 | 32.32 | 33.64 |
> > | Yi-1.5-9B-Chat-16K | 36.44 | 36.30 | 35.77 | 36.28 | 39.98 | 37.61 | 40.66 | 35.73 | 43.22 | 37.67 |
> > | Qwen3-8B | 36.39 | 35.27 | 36.72 | 35.48 | 37.02 | 35.59 | 39.90 | 35.91 | 40.20 | 36.49 |
> > | Analyst | 28.98 | 33.81 | 32.36 | 33.80 | 31.60 | 31.96 | 31.06 | 36.49 | 32.95 | 33.08 |

---

### Official Review · Reviewer_PycM · 2025-10-31

**Soundness:** 1
**Presentation:** 2
**Contribution:** 1
**Rating:** 2
**Confidence:** 5

**Summary:**

The paper introduces Fin-Herding, a benchmark designed to evaluate whether large language models exhibit herding behavior when exposed to human-biased financial opinions. The authors compare LLM-generated investment ratings with analyst ratings and actual stock returns. They find that when analyst ratings are visible within the reports, LLMs’ alignment with analysts increases significantly. However, masking analyst ratings reduces this effect, and some models even outperform analysts in predicting future stock performance under these conditions.

**Strengths:**

The paper documents a new phenomenon, “LLM herding,” in the financial context. The authors perform extensive empirical analyses using a host of different LLMs, confirming the existence of herding across different model classes. They also provide a benchmark dataset that could be potentially useful for future research.

**Weaknesses:**

The authors’ claim that LLMs are susceptible to herding when presented with analysts’ ratings is not particularly surprising. Furthermore, there are several critical issues with the research design.

1.	LLM prompts do not change the model weights but change the model’s answering patterns. Analysts’ ratings are arguably the most important information that effectively summarizes the overall document. Even though it constitutes a very small part of the entire text, the rating itself will arguably receive significant attention when the models process the text. Given this, it is not very surprising that the models place an emphasis on the summarized rating and follow the information. To empirically assess whether the ratings of analyst reports attract disproportionate model attention, the authors could implement an open-source experiment using models. Specifically, they can compute each layer–head’s attention share directed toward the rating tokens, identified via phrases like “Buy,” “Hold,” or “Sell,” relative to all other tokens in the text. Comparing this “attention share to rating” with control spans of equivalent length located elsewhere in the report would reveal whether attention mass systematically concentrates on the rating. Furthermore, I am confused about what the readers can learn from this seemingly evident observation. The authors should clarify how this research can contribute both to academics and practitioners. The authors claim that they “propose a potential way to avoid herding.” Are they referring to the “masking” that they performed?

2.	If the authors wanted to effectively test whether LLMs really do herd, they could have presented the model with “manipulated” ratings. That is, they could have changed the rating from bearish to bullish but retained the remaining content. If the models ignore the content and simply follow the overall ratings, the authors could argue that the models are heavily affected by the ratings themselves. In any case, I believe that this counterfactual experiment is required.

3.	Another interesting experiment aligning more closely with the economic definition of “herding” would be to present multiple LLM agents with a company-related newspaper article and ask each to independently form stock return expectations. In a follow-up experiment, allowing the agents to “discuss” or exchange outputs before re-estimating their expectations would reveal whether their forecasts converge. A decrease in disagreement among the agents after the discussion would provide clear evidence of herding behavior among LLMs.

4.	The authors’ research design does not necessarily reflect how investors make use of LLMs when making investment decisions. Investors seldom use analyst reports to form price expectations. As the authors note, analyst reports are the most biased financial documents. MD&As in 10-Ks are biased but are arguably less biased than analyst reports. Investors, including quant firms, make use of LLM agents to analyze lengthy financial documents and extract key information. They then make decisions based on the LLM outputs (either using professional judgment or quantitative tools on the generated outputs). I recommend that the authors explore other “less biased” financial texts than analyst reports. It is not very surprising that the model becomes more biased when the text itself is heavily biased with opinions. Several alternatives that I could suggest are: (i) MD&A texts from 10-Ks (less biased than analyst reports), and (ii) financial statements and footnotes from 10-Ks (largely boilerplate yet neutral).

5.	Another interesting qualitative exercise would be to investigate where LLMs generate different ratings from analysts’ ratings when ratings are masked. It is important to understand why LLMs made such deviations. Do the reports themselves contain more negative information than positive information but end up providing a buy recommendation (or vice versa)?

6.	Additionally, the stock prediction results reported in Tables 5 and 6 are concerning. The accuracies are below 50%, implying that one could potentially earn positive alpha by taking the opposite position of the models’ predictions. This contradicts prior research showing that LLMs can achieve above-random accuracy in predicting returns. The discrepancy likely arises because the analyst reports used as input are already biased, even when explicit analyst ratings are masked. Hence, the authors should tone down claims that LLMs “outperform analysts” and interpret the 40–50% predictive accuracy more cautiously, clarifying that it may reflect systematic bias in the source documents rather than true model capabilities.

**Questions:**

Please see above.

---

> ### Author Response · Authors · 2025-11-27
> **Thanks for your careful review and we have made major revisions based on your comments**
>
> Following your suggestions on “If the authors wanted to effectively test whether LLMs really do herd, they could have presented the model with “manipulated” ratings. “ , we add another experiment. Following your suggestions, we introduce a fabricated analyst rating by deliberately altering the rating stated in the first sentence of the original report. This enables us to assess the extent to which LLMs adopt—or “herd toward”—a false rating even when it contradicts the subsequent analysis. The results suggest that models have different degrees of “herding” rather than alignment. Even for GPT-4, there is a near 50% percentage of samples providing investment rating herding towards fake rating. The detail results are as follows:
>
> **Table: Model Herding with 'fake' rating**
>
> | Model | Comm. Services | Cons. Defensive | Energy | Financial Services | Healthcare | Industrials | Real Estate | Technology | Utilities | Avg |
> |-------|----------------|-----------------|--------|---------------------|------------|-------------|-------------|------------|-----------|------|
> | **Private Models** |||||||||||
> | GPT-5 | 50.00 | 39.61 | 32.20 | 26.70 | 28.49 | 29.80 | 38.64 | 32.19 | 42.65 | 33.54 |
> | GPT-4 | 64.24 | 41.55 | 60.65 | 58.38 | 51.24 | 43.87 | 57.07 | 30.71 | 55.92 | 48.77 |
> | Claude-3.5-Haiku | 60.23 | 39.34 | 58.65 | 54.97 | 48.12 | 46.03 | 54.62 | 36.41 | 50.71 | 45.98 |
> | Claude-4-Sonnet | 52.06 | 40.60 | 37.09 | 30.75 | 33.40 | 43.73 | 35.45 | 36.70 | 40.02 | 35.85 |
> | **Open-source Models** |||||||||||
> | Qwen2.5-7B-it | 41.33 | 22.96 | 41.19 | 33.96 | 33.01 | 34.07 | 32.96 | 14.77 | 34.08 | 30.33 |
> | Qwen2-7B-it | 52.65 | 28.47 | 42.29 | 40.66 | 31.10 | 22.02 | 36.48 | 20.57 | 33.28 | 31.69 |
> | Qwen2.5-14B-it | 65.20 | 38.89 | 60.79 | 62.80 | 50.56 | 33.66 | 54.55 | 35.17 | 40.00 | 46.64 |
> | Qwen3-8B | 46.18 | 31.53 | 64.72 | 45.79 | 44.95 | 26.17 | 45.57 | 17.43 | 38.25 | 36.89 |
> | gemma-2-27b-it | 68.76 | 43.25 | 67.64 | 62.00 | 53.81 | 42.20 | 58.22 | 23.07 | 55.51 | 48.78 |
> | gemma-2-9b-it | 48.01 | 19.68 | 49.25 | 29.15 | 30.06 | 16.61 | 33.18 | 17.74 | 31.39 | 27.14 |
> | gemma-7b-it | 19.93 | 11.62 | 20.98 | 7.91 | 9.95 | 12.59 | 9.34 | 7.57 | 12.98 | 11.48 |
> | internlm2-chat-7b | 39.91 | 30.46 | 39.55 | 35.28 | 29.70 | 25.08 | 29.55 | 25.23 | 26.25 | 30.34 |
> | Mistral-7B-it-v0.3 | 69.90 | 59.04 | 72.96 | 69.77 | 68.35 | 55.24 | 65.15 | 37.74 | 66.38 | 60.42 |
> | Mistral-Nemo-it-2407 | 50.17 | 30.00 | 40.52 | 37.81 | 29.03 | 26.50 | 30.96 | 22.45 | 29.18 | 31.28 |
> | Yi-1.5-9B-Chat-16K | 32.43 | 27.24 | 38.50 | 34.63 | 33.80 | 32.40 | 34.01 | 30.85 | 34.91 | 32.80 |
> | DeepSeek-V2-Lite-Chat | 57.09 | 56.80 | 55.54 | 60.69 | 56.60 | 55.52 | 58.87 | 47.52 | 59.86 | 55.87 |
> | Meta-Llama-3-8B-it | 64.40 | 42.78 | 50.41 | 47.95 | 48.37 | 39.40 | 46.95 | 56.06 | 47.28 | 48.57 |
> | Meta-Llama-3.1-8B-it | 34.80 | 17.60 | 26.63 | 19.35 | 19.17 | 12.08 | 22.78 | 14.12 | 19.66 | 18.78 |
> | glm-4-9b-chat | 70.99 | 46.61 | 62.56 | 53.73 | 53.65 | 45.29 | 51.91 | 39.88 | 58.56 | 51.30 |

---

> > ### Author Response · Authors · 2025-11-27
> >
> > Regarding your other comments, some are out of scope for our paper. Our main focus is comprehensive evaluation of LLM herding, like how significant is herding, which models tend to herd and some potential solutions. Our work is to shed a light for future research. Your suggestions like “ present multiple LLM agents, investigate where LLMs generate different ratings from analysts’ ratings when ratings are masked” can potentially form another paper. If we further add those things, the paper would be lengthy. This is CS conference, not ECON/ACCT/FIN  journal. How can 10 pages accommodate that many extensions? I suggest you to read some benchmark/dataset papers that published in top cs conferences.
> >
> > Since our goal to evaluate a broad range LLMs including private and open-source, using MD&A is impossible since MD&A on average has 30,000 tokens which is far beyond the maximum context length of most open-source LLMs. We select a broad range of open-source LLMs that have 8192 maximum context length. Therefore, only a carefully summarized analyst report is suitable for this task.
> >
> > Finally, we do not quite understand your statements “The accuracies are below 50%, implying that one could potentially earn positive alpha by taking the opposite position of the models’ predictions. This contradicts prior research showing that LLMs can achieve above-random accuracy in predicting returns. “ Since our true label has three categories ‘bullish’, ‘neutral’ and ‘bearish’, it is not like simply predicting ‘increase’ or ‘decrease’, so why the random guess accuracy is 50% ???

---

### Official Review · Reviewer_qnuF · 2025-11-02

**Soundness:** 1
**Presentation:** 2
**Contribution:** 1
**Rating:** 2
**Confidence:** 4

**Summary:**

This paper introduces Fin-Herding, a benchmark designed to evaluate whether large language models (LLMs) exhibit herding behavior, defined here as alignment with human (analyst) opinions, when making financial investment judgments. Using 8,868 analyst reports sourced from Yahoo Finance, the authors construct two versions of each report (with and without analyst rating) and prompt 18 open- and closed-source LLMs to output their own bullish/neutral/bearish investment ratings. The proposed herding score measures the degree of agreement between model and analyst ratings. Empirically, when analyst opinions are included, all models’ herding scores rise by roughly 5–10%, suggesting that the presence of human ratings drives greater conformity. The authors interpret this as evidence that LLMs “follow human bias” and lose independent reasoning capacity.

**Strengths:**

1. Cross-model and cross-sector evaluation: The experiments span 18 LLMs (including GPT-5, Claude-4, Llama-3, Gemma, Qwen2, etc.) and multiple industries, offering broad empirical coverage.
2. Interesting motivation:  Exploring how LLMs integrate, or over-rely on, human judgments in finance touches an important area related to model alignment, trust, and bias propagation.

**Weaknesses:**

1. Conceptual Ambiguity: “Herding” is defined purely as label agreement between LLM and analyst, without distinguishing between rational information uptake and irrational conformity. In behavioral finance, herding implies unjustified convergence of opinions despite private signals. Here, however, concordance could simply reflect legitimate reasoning based on shared evidence. Without counterfactual or adversarial setups (e.g., feeding incorrect analyst ratings), the observed effect cannot be interpreted as herding in the financial sense.

2. Misinterpretation of Context Sensitivity: The finding that LLMs’ outputs shift when given additional contextual information is unsurprising. It merely shows standard context conditioning or sycophancy, a well-known phenomenon already analyzed in general-domain studies such as Sharma et al. (ICLR 2024) and Xie et al. (ICLR 2024). The claim that this behavior reveals a unique financial herding bias therefore overstates novelty.

3. Experimental Design and Statistical Validity:
- Severe label imbalance. Only 0.29% of analyst samples are “bearish” while over 72% are bullish, yet all evaluations rely on plain accuracy. No class weighting, resampling, or calibration metrics are applied, making comparisons statistically unreliable.
- Arbitrary “true label.” The “ground truth” based on one-month return ±1% is unjustified. Analyst reports typically express 6–12-month horizons, and monthly stock returns are dominated by market noise. This weakens any inference about predictive accuracy or “analyst vs. model” performance.
- Incomplete control for implicit sentiment cues. Only the first sentence (containing the rating) is masked; stylistic or lexical cues throughout the report can still leak sentiment, making the masked/unmasked distinction unreliable.
- No statistical testing. Differences of 5–10% in accuracy or herding score are reported without significance testing or confidence intervals.

**Questions:**

1. What is the conceptual distinction between LLM alignment and herding, and how should they be properly differentiated?

2. From an experimental perspective, several questions remain — in particular, what is the rationale for defining the ground truth based on a one-month return with a ±1% threshold?

---

> ### Author Response · Authors · 2025-11-27
> **Thanks for your careful review and we have made major revisions based on your comments.**
>
> First, regarding ‘herding conceptual ambiguity’, we add another experiment. Following your suggestions, we introduce a fabricated analyst rating by deliberately altering the rating stated in the first sentence of the original report. This enables us to assess the extent to which LLMs adopt—or “herd toward”—a false rating even when it contradicts the subsequent analysis. The results suggest that models have different degrees of “herding” rather than alignment. Even for GPT-4, there is a near 50% percentage of samples providing investment rating herding towards fake rating. The detail results are as follows:
>
> **Table: Model Herding with 'fake' rating**
>
> | Model | Comm. Services | Cons. Defensive | Energy | Financial Services | Healthcare | Industrials | Real Estate | Technology | Utilities | Avg |
> |-------|----------------|-----------------|--------|---------------------|------------|-------------|-------------|------------|-----------|------|
> | **Private Models** |||||||||||
> | GPT-5 | 50.00 | 39.61 | 32.20 | 26.70 | 28.49 | 29.80 | 38.64 | 32.19 | 42.65 | 33.54 |
> | GPT-4 | 64.24 | 41.55 | 60.65 | 58.38 | 51.24 | 43.87 | 57.07 | 30.71 | 55.92 | 48.77 |
> | Claude-3.5-Haiku | 60.23 | 39.34 | 58.65 | 54.97 | 48.12 | 46.03 | 54.62 | 36.41 | 50.71 | 45.98 |
> | Claude-4-Sonnet | 52.06 | 40.60 | 37.09 | 30.75 | 33.40 | 43.73 | 35.45 | 36.70 | 40.02 | 35.85 |
> | **Open-source Models** |||||||||||
> | Qwen2.5-7B-it | 41.33 | 22.96 | 41.19 | 33.96 | 33.01 | 34.07 | 32.96 | 14.77 | 34.08 | 30.33 |
> | Qwen2-7B-it | 52.65 | 28.47 | 42.29 | 40.66 | 31.10 | 22.02 | 36.48 | 20.57 | 33.28 | 31.69 |
> | Qwen2.5-14B-it | 65.20 | 38.89 | 60.79 | 62.80 | 50.56 | 33.66 | 54.55 | 35.17 | 40.00 | 46.64 |
> | Qwen3-8B | 46.18 | 31.53 | 64.72 | 45.79 | 44.95 | 26.17 | 45.57 | 17.43 | 38.25 | 36.89 |
> | gemma-2-27b-it | 68.76 | 43.25 | 67.64 | 62.00 | 53.81 | 42.20 | 58.22 | 23.07 | 55.51 | 48.78 |
> | gemma-2-9b-it | 48.01 | 19.68 | 49.25 | 29.15 | 30.06 | 16.61 | 33.18 | 17.74 | 31.39 | 27.14 |
> | gemma-7b-it | 19.93 | 11.62 | 20.98 | 7.91 | 9.95 | 12.59 | 9.34 | 7.57 | 12.98 | 11.48 |
> | internlm2-chat-7b | 39.91 | 30.46 | 39.55 | 35.28 | 29.70 | 25.08 | 29.55 | 25.23 | 26.25 | 30.34 |
> | Mistral-7B-it-v0.3 | 69.90 | 59.04 | 72.96 | 69.77 | 68.35 | 55.24 | 65.15 | 37.74 | 66.38 | 60.42 |
> | Mistral-Nemo-it-2407 | 50.17 | 30.00 | 40.52 | 37.81 | 29.03 | 26.50 | 30.96 | 22.45 | 29.18 | 31.28 |
> | Yi-1.5-9B-Chat-16K | 32.43 | 27.24 | 38.50 | 34.63 | 33.80 | 32.40 | 34.01 | 30.85 | 34.91 | 32.80 |
> | DeepSeek-V2-Lite-Chat | 57.09 | 56.80 | 55.54 | 60.69 | 56.60 | 55.52 | 58.87 | 47.52 | 59.86 | 55.87 |
> | Meta-Llama-3-8B-it | 64.40 | 42.78 | 50.41 | 47.95 | 48.37 | 39.40 | 46.95 | 56.06 | 47.28 | 48.57 |
> | Meta-Llama-3.1-8B-it | 34.80 | 17.60 | 26.63 | 19.35 | 19.17 | 12.08 | 22.78 | 14.12 | 19.66 | 18.78 |
> | glm-4-9b-chat | 70.99 | 46.61 | 62.56 | 53.73 | 53.65 | 45.29 | 51.91 | 39.88 | 58.56 | 51.30 |

---

> > ### Author Response · Authors · 2025-11-27
> >
> > Second, regarding label imbalance and arbitrary true label, it is difficult to get true label in the finance domain due to different investor horizons. Most prior literature (liu et al.(2023), Yu et al.(2024) use arbitrary labels. To address this issue, we develop a fine-grained approach for constructing ground-truth investment ratings using portfolio-based alpha generation standard. Specifically, we employ a quantile-based portfolio classification derived from the three-month (approximately 60 trading days) cumulative abnormal return (CAR) following the report issuance date. This design reflects the fact that analyst reports primarily target a firm’s medium- to long-term performance rather than short-horizon fluctuations in daily returns. On average, analysts update their forecasts when firm issuing quarterly report, so we believe 60 trading days is average forecast horizons. In particular, for each firm, given a report issued on YYYY/MM/DD, we compute the 60-day abnormal return ($\alpha$) using the market model. As shown in Equation (1), we regress firm-level returns on market returns to estimate the risk-adjusted benchmark ($\beta$), and the resulting residual captures the abnormal performance (alpha). Because investors such as hedge funds are fundamentally concerned with risk-adjusted abnormal returns rather than raw buy-and-hold performance, cumulative abnormal return provides a more economically meaningful basis for defining true investment ratings. After computing each report’s 60-day cumulative abnormal return, we classify investment outcomes using a year-specific, quantile-based approach inspired by long-short portfolio method widely documented in finance literature. For all reports issued in year t, we sort the post-report cumulative abnormal returns and assign ratings based on their relative positions within the annual distribution. Specifically, observations in the upper 30\% quantile are labeled bullish, those in the lower 30\% quantile are labeled bearish, and the remaining middle 40\% are categorized as neutral. This procedure ensures that classifications are comparable across years and are not distorted by time-varying market conditions. The new distribution of the true label is shown as below, which is more balanced. The analysts have way more ‘bullish’ rating than ‘bearish’ rating is the FACT, and that biased estimate is what we are studying. So we don’t think we need to do class reweighting based on analyst rating which may lead to a big loss of sample. The new results suggest that some model investment rating accuracies deviate a lot without herding analysts using new true label. The details can also be found in the updated version.
> >
> > Table: Sample Investment Rating {#tab:SampleAnalysts}
> >
> > | Analyst Investment Rating | Num  | Percentage |
> > |---------------------------|------|------------|
> > | Bullish                   | 6410 | 72.28%     |
> > | Bearish                   | 26   | 0.29%      |
> > | Neutral                   | 2432 | 24.74%     |
> > | **Total**                 | 8868 | 100%       |
> >
> > **Return-based Rating**
> >
> > | Return-based Rating | Num  | Percentage |
> > |---------------------|------|------------|
> > | Bullish             | 2664 | 30%        |
> > | Bearish             | 2662 | 30%        |
> > | Neutral             | 3542 | 40%        |

---

> > > ### Author Response · Authors · 2025-11-27
> > >
> > > Third, regarding “implicit sentiment cues”, we need to emphasize that our main aim is to evaluate the degree of model herding even when a very small part of opinions (one sentence) change in a long context. At the same time, to identify all “implicit sentiment cues”, we employ the Multi-Perspective Question Answering (MPQA) Subjectivity Lexicon, a widely used linguistic resource for detecting opinion-bearing and subjective expressions in text. The lexicon comprises several thousand lexical items annotated with rich subjectivity metadata—including polarity (positive, negative, neutral, or both), subjectivity strength (“strongsubj” or “weaksubj”), and part-of-speech categories (noun, verb, adjective, adverb). In our implementation, we perturb each analyst report by removing sentences that contain any lexical item labeled as strongsubj in MPQA and input to LLMs. As reported in the following table, the method proves effective for both private models and open-source models like Qwen-3 8B, GPT-4 with 2% - 4% gains. Most models perform even better than analysts.
> > >
> > > **Table: Model Accuracy without human opinions**
> > >
> > > | Model | Comm. Services | Cons. Defensive | Energy | Financial Services | Healthcare | Industrials | Real Estate | Technology | Utilities | Avg |
> > > |-------|----------------|-----------------|--------|---------------------|------------|-------------|-------------|------------|-----------|------|
> > > | **Private Models** |||||||||||
> > > | GPT-5 | 33.56 | 37.78 | 34.86 | 32.92 | 31.52 | 33.91 | 33.59 | 34.74 | 35.66 | 34.16 |
> > > | GPT-4 | 36.27 | 36.56 | 33.82 | 34.38 | 35.87 | 35.65 | 36.11 | 36.01 | 39.37 | 35.88 |
> > > | Claude-3.5-Haiku | 35.31 | 37.12 | 35.04 | 33.43 | 30.98 | 33.74 | 35.25 | 35.03 | 34.61 | 34.99 |
> > > | Claude-4-Sonnet | 35.79 | 37.54 | 33.96 | 35.13 | 34.69 | 35.63 | 36.86 | 35.47 | 38.98 | 35.70 |
> > > | **Open-source Models** |||||||||||
> > > | Qwen2-7B-It | 32.63 | 33.12 | 34.79 | 33.03 | 31.84 | 33.12 | 38.30 | 36.57 | 33.14 | 33.82 |
> > > | gemma-2-9b-it | 38.11 | 35.77 | 34.27 | 34.52 | 33.13 | 32.68 | 35.78 | 36.03 | 35.35 | 34.71 |
> > > | gemma-7b-it | 33.56 | 31.87 | 32.20 | 35.04 | 31.08 | 32.17 | 35.86 | 35.77 | 29.81 | 33.12 |
> > > | glm-4-9b-chat | 33.28 | 31.27 | 34.81 | 32.45 | 30.42 | 31.50 | 35.03 | 35.13 | 30.55 | 32.53 |
> > > | Mistral-Nemo-It-2407 | 36.50 | 33.50 | 33.17 | 32.74 | 33.83 | 34.64 | 36.80 | 35.89 | 36.29 | 34.58 |
> > > | Mistral-7B-It-v0.3 | 38.81 | 34.05 | 36.64 | 35.26 | 37.89 | 34.84 | 38.38 | 35.33 | 36.23 | 35.99 |
> > > | DeepSeek-V2-Lite-Chat | 37.09 | 32.17 | 35.07 | 33.26 | 36.29 | 38.06 | 35.04 | 35.20 | 35.24 | 35.30 |
> > > | Meta-Llama-3-8B-It | 36.56 | 35.20 | 35.02 | 35.19 | 33.33 | 33.87 | 38.07 | 34.81 | 35.71 | 34.91 |
> > > | Meta-Llama-3.1-8B-It | 33.05 | 32.24 | 33.82 | 33.04 | 31.05 | 31.48 | 32.06 | 33.71 | 33.43 | 32.58 |
> > > | internlm2-chat-7b | 36.01 | 32.76 | 36.54 | 32.54 | 32.70 | 33.10 | 34.49 | 35.25 | 32.09 | 33.69 |
> > > | gemma-2-27b-it | 36.94 | 35.97 | 35.66 | 35.73 | 35.36 | 35.09 | 42.09 | 36.96 | 36.52 | 36.23 |
> > > | Qwen2.5-14B-It | 32.59 | 33.89 | 34.28 | 35.92 | 35.37 | 32.17 | 38.46 | 35.63 | 33.67 | 34.60 |
> > > | Qwen2.5-7B-It | 33.59 | 31.77 | 34.19 | 33.66 | 33.39 | 33.49 | 36.13 | 34.93 | 32.32 | 33.64 |
> > > | Yi-1.5-9B-Chat-16K | 36.44 | 36.30 | 35.77 | 36.28 | 39.98 | 37.61 | 40.66 | 35.73 | 43.22 | 37.67 |
> > > | Qwen3-8B | 36.39 | 35.27 | 36.72 | 35.48 | 37.02 | 35.59 | 39.90 | 35.91 | 40.20 | 36.49 |
> > > | Analyst | 28.98 | 33.81 | 32.36 | 33.80 | 31.60 | 31.96 | 31.06 | 36.49 | 32.95 | 33.08 |
> > >
> > > Lastly, we do see some prior literature also document context conditioning or sycophancy, but they use short-form, factual-based question-answering in general domain, and they focus on model training like using RLHF. In contrast, our research focuses on model inference in a specific finance task and the input is long-context-based. Also, we propose methods to reduce the side effects(herding) of model training in the finance area and show how to better use LLM, which is AI for Science chasing for.  Therefore, we think our work is still meaningful and we sincerely hope our revisions can address your concerns.

---

### Note · Authors · 2026-01-06

I have read and agree with the venue's withdrawal policy on behalf of myself and my co-authors.